# Time to Step Up Conservation: Climate Change Will Further Reduce the Suitable Habitats for the Vulnerable Species Marbled Polecat (*Vormela peregusna*)

**DOI:** 10.3390/ani13142341

**Published:** 2023-07-18

**Authors:** Xiaotian Cheng, Yamin Han, Jun Lin, Fan Jiang, Qi Cai, Yong Shi, Dongyang Cui, Xuanye Wen

**Affiliations:** 1The Station of Forest Seedling Quarantine and Pest Management, Changji 831100, China; chengxiaotian8862@163.com (X.C.); hym0727@126.com (Y.H.); 2Locust and Rodent Control Headquarters of Xinjiang Uygur Autonomous Region, Urumqi 830000, China; jun_lin2013@163.com; 3Center for Biological Disaster Prevention and Control, National Forestry and Grassland Administration, Shenyang 110031, China; jiangfan930430@163.com (F.J.); yshisyau@163.com (Y.S.); cuidy@bdpc.org.cn (D.C.); 4Institute of Ecological Conservation and Restoration, Chinese Academy of Forestry, Beijing 100091, China; caiqilinda1010@126.com

**Keywords:** *Vormela peregusna*, suitable area, red list, species distribution models, Biomod2

## Abstract

**Simple Summary:**

Assessing suitable habitats for species has great potential to guide the management and conservation of threatened species, especially rare species that are poorly studied and remote from human populations. *Vormela peregusna*, a mustelid mammal endemic to Eurasia, was assessed as vulnerable by the International Union for Conservation of Nature in 2015 due to its low population size and increasing human threats. We used the ensemble model to assess the distribution of suitable areas for *V. peregusna* under current and future climate change conditions to contribute to the better protection of endangered animals.

**Abstract:**

Habitat loss and human threats are putting the marbled polecat (*Vormela peregusna)* on the brink of extinction. Numerous recent studies have found that climate change will further deteriorate the living environment of endangered species, leading to their eventual extinction. In this study, we used the results of infrared camera surveys in China and worldwide distribution data to construct an ensemble model consisting of 10 commonly used ecological niche models to specify potential suitable habitat areas for *V. peregusna* under current conditions with similar environments to the sighting record sites. Changes in the suitable habitat for *V. peregusna* under future climate change scenarios were simulated using mid-century (2050s) and the end of the century (2090s) climate scenarios provided by the Coupled Model Intercomparison Project Phase 6 (CMIP6). We evaluated the accuracy of the model to obtain the environmental probability values (cutoff) of the *V. peregusna* distribution, the current distribution of suitable habitats, and future changes in moderately and highly suitable habitat areas. The results showed that the general linear model (GLM) was the best single model for predicting suitable habitats for *V. peregusna*, and the kappa coefficient, area under the curve (AUC), and true skill statistic (TSS) of the ensemble model all exceeded 0.9, reflecting greater accuracy and stability than single models. Under the current conditions, the area of suitable habitat for *V. peregusna* reached 3935.92 × 10^4^ km^2^, suggesting a wide distribution range. In the future, climate change is predicted to severely affect the distribution of *V. peregusna* and substantially reduce the area of suitable habitats for the species, with 11.91 to 33.55% of moderately and highly suitable habitat areas no longer suitable for the survival of *V. peregusna*. This shift poses an extremely serious challenge to the conservation of this species. We suggest that attention be given to this problem in Europe, especially the countries surrounding the Black Sea, Asia, China, and Mongolia, and that measures be taken, such as regular monitoring and designating protected areas for the conservation of vulnerable animals.

## 1. Introduction

Determining how to protect endangered species is one of the main problems faced in conservation biology [1,2]. The most recent evaluation showed that over 42,100 species are at risk of extinction to varying degrees, a number that amounts to 28% of all species assessed [3]. Habitat loss and fragmentation are among the most important issues currently observed. Understanding habitat suitability and the factors that influence species habitats is the basis for the protection of threatened species [4,5]. In general, the risk of warming due to climate change will exacerbate the loss of species habitat [6]. Additionally, the loss of fauna may be imminent if scientifically sound conservation measures are not developed, especially for animals that are already scarce.

Habitat analysis is an important prerequisite for conservation planning and managerial decision making [7] and is urgently needed to enhance the conservation of endangered species. Species distribution models (SDMs) use environmental variables associated with species distribution sites to predict the ecological needs and potential distribution of species and are widely used in invasion biology, conservation biology, global change biology, and risk of disease transmission [8]. Similar to other ecological models, ecological niche models have uncertainty in their predictions, which are closely related to species distribution sites, environmental variables, and model algorithms and parameters, among which the species distribution sites and model algorithms have the greatest effects [9]. There are more than 30 SDMs available [10], each with different advantages; however, it is difficult to choose the optimal assessment model for species that have rarely been studied [11]. Biomod2 is an ensemble model based on a variety of single models [12] and provides better prediction accuracy and spatial sensitivity to small sample sets than do single models, which are prone to overfitting and uncertainty [13,14,15].

*Vormela peregusna* is the only small mammal in the genus *Vormela* of the order Carnivora of the class Mammalia [16]. *V. peregusna* is valuable for ecosystem stability and medical research. On the one hand, it is a major predator of desert rodents [17] and can effectively protect desert vegetation from rodent damage; on the other hand, *V. peregusna* is associated with many tick-borne pathogens [18], and research on this species has helped constrain the transmission mechanisms of zoonotic diseases. Unfortunately, despite the value of this species, the risk to the survival of *V. peregusna* is very concerning. As early as 2008, *V. peregusna* was included on the Red List of Threatened Species by the International Union for Conservation of Nature (IUCN), and a subsequent reassessment found that the *V. peregusna* population declined by 30% in a decade [19], reaching a classification of vulnerable. Although China, Bulgaria [20], and other countries have classified *V. peregusna* as a protected species, thus far, no countries have established targeted conservation measures to save *V. peregusna* from extinction.

We hypothesize that there is a direct relationship between the species distribution of *V. peregusna* and its food sources, and since there are no efficient methods to monitor *V. peregusna* populations at a broad scale, currently, most records come from opportunistic sightings [21]. Observations from southern Europe show that *V. peregusna* prefers to appear in farmland–grassland interlaced areas with good vegetation conditions [22]. However, this situation is completely different in Asia and the Middle East, with records of occurrences in countries such as Iraq, China, and Mongolia coming from sparsely vegetated hilly and desert areas [23,24]. Another strong piece of evidence is that, in infrared camera monitoring of *V. peregusna*’s main food *Rhombomys opimus*, it was found that the activity rhythm of *V. peregusna* was positively correlated with that of *R. opimus* [25]. When the activity of *R. opimus* increased or decreased with seasonal changes, the activity pattern of *V. peregusna* also changed accordingly. Although the effect of food on the survival of *V. peregusna* has not yet been clearly determined, it will be of great help to protect *V. peregusna* if protected areas can be delineated by considering the suitable range and occurrence of rodents.

Habitat loss is the main reason for the endangerment of *V. peregusna*. In Europe, the conversion of much grassland to cropland has reduced the suitable habitat for *V. peregusna*, while in Asia, land desertification is the main threat to *V. peregusna* [26]. Several past studies have found that climate change will have a dramatic impact on the stability of desert and grassland ecosystems [27,28]. Studies from Central Asian grasslands have shown that changes in precipitation are the main factor causing changes in grassland vegetation [29], and studies on the survival of desert fauna have also shown a significant decline in desert bird populations over the past century due to temperature and precipitation [30]. It is also important to address whether the habitat of *V. peregusna*, as a desert and grassland habitat species, will be affected by climate change.

In this study, we collected monitoring data obtained with infrared cameras in the desert areas of northwest China and reports of *V. peregusna* distribution worldwide to improve the accuracy of model-based distribution predictions. Ten single models were established based on Biomod2, and the ability of different models to predict suitable habitat areas for *V. peregusna* was compared. An ensemble model was used to predict the distribution of *V. peregusna* and changes in suitable habitat areas in current and future situations. Specifically, our aims were to (a) provide new strategies for conducting suitable habitat studies of species with small populations, (b) enhance the conservation of vulnerable animals, and (c) recommend conservation priority areas for effective conservation in the future.

## 2. Materials and Methods

### 2.1. Occurrence Data

In previous studies, we used infrared cameras to monitor rodent pests in desert areas of China, and 12 *V. peregusna* distribution points were identified [25]. Additionally, a total of 491 distribution points was obtained by searching the database of the Global Biodiversity Information Facility (http://www.gbif.org, accessed on 18 October 2022) and research articles related to the distribution of *V. peregusna* [31,32,33,34]. Due to the duplication of many point co-ordinates, we removed redundant data to reduce the error caused by the clustering effect so that only one distribution point was retained in each grid (100 km^2^). Ultimately, 101 valid points were obtained (Figure 1), and the latitude and longitude co-ordinates of each point are shown in Appendix A.

### 2.2. Selection and Processing of Environmental Variables

We used 31 environmental variables for modeling, and these environmental variables were shown to be directly related to *R. opimus* distribution in our earlier modeling [35]: 19 bioclimatic factors, 9 soil factors, and 3 topographic factors. The climate data were downloaded from the WorldClim database (http://worldclim.org, accessed on 19 October 2022), and the future climate scenarios were downloaded from the BCC-CSM (Beijing Climate Center, China Meteorological Administration, Beijing, China). These scenarios included those with low (SSP126), moderate (SSP245), and high (SSP585) emissions of greenhouse gases [36]. The soil and topographic factor data were obtained from the Harmonized World Soil Database (HWSD) of the Food and Agriculture Organization of the United Nations (http://www.fao.org/faostat/en/#data, accessed on 19 October 2022), with the spatial resolution of each factor set to 10 arc min [37].

The interactions between environmental factors can lead to collinearity issues during modeling and subsequent overfitting [38]. First, variance inflation factor (VIF) analysis was used to select the most important environmental variables. Second, R was used to conduct a Pearson correlation analysis, reduce the complexity of the model, and improve its prediction accuracy. Factors with a correlation less than 0.8 were preliminarily selected, and, from those, factors with a VIF less than 10 were retained. Then, based on the Pearson correlation test results, factors with a correlation coefficient less than 0.8 were retained, and factors with correlation coefficients greater than 0.8 were omitted. A total of 18 environmental variables were selected (Table 1).

### 2.3. Model Construction

To reduce the modeling bias caused by the uncertainty inherent in models, we first performed fitting with 10 single methods included in Biomod2: a generalized additive model (GAM), a generalized boosted regression model (GBM), a general linear model (GLM), a random forest (RF), the multivariate adaptive regression splines (MARS) method, classification tree analysis (CTA), an artificial neural network (ANN), the surface range envelope (SRE) method, flexible discriminant analysis (FDA), and the maximum entropy (Maxent) method. Before constructing the model, it was necessary to process the species distribution data. Biomod2 provides several methods to generate nonexistence (pseudoabsence) points from background research data [39]. The “random” command was used to randomly generate 1200 pseudoabsence data points for model simulation. Then, the “biomod_tuning” command was used to optimize the model parameters and select 70% of the sample data for training. The remaining 30% of the sample data were used to verify the performance of the model [40]. The resulting single models were evaluated using 3 metrics: the true skill statistic (TSS), AUC, and kappa coefficient [41].

Single models with accuracies that met the selected standard were integrated into an ensemble model using a weighted average approach [42]. First, the results of the single models used in the construction of the ensemble model were normalized so that the predictions of single SDMs were in the range of [0, 1]. This process was repeated 10 times to avoid random errors associated with the use of a single model. Then, the weights for model combination were determined based on the AUC and TSS values of each model, and the single models used to construct the ensemble model were determined with fixed cutoffs of TSS > 0.7 and AUC > 0.8. The higher the average AUC and TSS values were after multiple runs, the greater the weight assigned to the corresponding single model was when it was incorporated into the ensemble model.

### 2.4. Changes in the Spatial Pattern of the Suitable Distribution Ranges of Species

The 0/1 probability value cutoff of “suitable” or “unsuitable” was obtained by running the model. The spatial units with values below the cutoff were considered unsuitable habitats, and the spatial units with values above the cutoff were divided into 3 equal parts, corresponding to minimally, moderately, and highly suitable habitats [43]. Two time nodes, namely, the middle of this century and the end of this century, were selected to analyze the future suitable area changes of *V. peregusna*, and the average values from 2040–2060 (2050s) and 2080–2100 (2090s) were calculated [44]. Based on the “binary_meth” operation in Biomod2, we obtained the results of the suitable/unsuitable (0/1) simulation and used the “biomod_rangesize” function to calculate the changes in the spatial pattern of the suitable areas of *R. opimus* under future climate change scenarios [45]. Finally, the results, in matrix format, were loaded into ArcGIS v10.4.1 for visual representation.

## 3. Results

### 3.1. Model Accuracy

Among the 10 models evaluated, only the GAM failed to run successfully due to the difficulty in obtaining parameter values; the other 9 models were all run successfully, and a total of 90 sets of results were obtained. The different models were compared (Appendix A), and the GBM displayed the highest accuracy and the best stability for the three evaluation metrics, suggesting that it was the best choice for assessing suitable habitats for *V. peregusna* using a single model. Although higher scores were obtained for the FDA and RF models, they each yielded one data anomaly in the calculation process. Moreover, low scores were obtained for the other models, which failed to reach acceptable performance levels. From the 90 sets of results, we selected a total of 39 eligible models to construct the ensemble model. The final ensemble model yielded a kappa coefficient of 0.91, a TSS value of 0.94, and an AUC value of 0.96, indicating excellent results.

### 3.2. Current Distribution Range

Suitable habitats for *V. peregusna* were found over almost all of Eurasia, except in a few tropical areas in Southeast Asia. Based on the current climate scenario, the suitable habitats for *V. peregusna* covered 3935.92 × 10^4^ km^2^, of which moderately suitable habitats accounted for 2415.17 × 10^4^ km^2^, followed by low-suitability habitats (867.99 × 10^4^ km^2^). Highly suitable habitats accounted for the smallest area (652.76 × 10^4^ km^2^). The highly suitable habitats were mainly found in the following regions: (1) the plain area at the border of China and Mongolia in the east, (2) the plateau area from the Orkhon River to Khangai Mountain in the west–central part of Mongolia, (3) the area from 40 to 50° N from Bulgaria and Ukraine in the west to the Junger Basin in China and the Siberian Plain in Russia in the east, and (4) sporadic highly suitable habitats in Spain, Italy, Hungary, Poland, Lithuania, Latvia, and the far east of Russia (Figure 2).

### 3.3. Future Changes in Suitable Habitat Area

Based on the three future climate scenarios, the geographical distributions of moderately and highly suitable habitats for *V. peregusna* were predicted to decrease to varying degrees (Figure 3). Although the percentages of the increases and decreases differed in various scenarios, the decrease in the area of suitable habitats for *V. peregusna* was much larger than the corresponding increase in each scenario. The lost suitable habitats were mainly in Europe, and most areas from northern to southern Europe were predicted to no longer be suitable for *V. peregusna* survival, especially in several countries around the Black Sea, where many *V. peregusna* have been recorded. Increases and decreases in the area of suitable habitats were variable in other regions, with decreases occurring in the eastern part of Saudi Arabia, in the border area between northeastern China and Russia, and on the Western Siberian Plain. Regions with increases in the area of suitable habitats were concentrated around the Ural Mountains in Russia, in the central and southern parts of the Arabian Peninsula, in North China, and in central Xinjiang, China.

The predicted area changes in each scenario indicated that both SSP126 and SSP585 in the middle of this century will result in extensive losses of moderately and highly suitable habitats for *V. peregusna* (Table 2), with the lost area exceeding 900 × 10^4^ km^2^, accounting for 30% of the area of existing suitable habitats. In comparison, the case of SSP245 was relatively optimistic, but the lost area still reached 646 × 10^4^ km^2^, which was 21.06% less than that in the current period. At the end of this century, SSP245 is projected to be the scenario with the most severe decrease in the area of suitable habitats for *V. peregusna*, and the species range is predicted to decrease to only two-thirds of the existing distribution area, with a loss of 35.83% of suitable habitat. SSP126 is the most optimistic scenario for habitat suitability based on the predicted results. Notably, the percentages of area gain and loss for *V. peregusna* habitats are projected to be 5.89% and 11.91%, respectively, with an overall change of only 11.91%. The change in the area of suitable habitat under SSP585 is predicted to be stable at the end of this century, and the increases and decreases in suitable habitat area are expected to be consistent with those at 50° S.

## 4. Discussion

In general, species ecological niches evolve at a much slower rate than climate change [46], and species respond to rapid climate change by dispersing to new suitable habitats, adapting, or becoming extinct [47]. Despite the widespread distribution of *V. peregusna* in Eurasia, it is alarming that our assessment indicated that a 12 to 34% decrease in the suitable habitats for *V. peregusna* may occur in the coming decades, implying that the survival of *V. peregusna* may be severely affected by climate change; this trend is largely associated with the extremely low number of *V. peregusna* per unit area of distribution and the poor migratory capacity of the species [19]. However, the future predictions obtained with the model in this study are relatively uncertain. Our results only indicate the probability of potential occurrence and do not represent real changes in the species distribution or habitat area [48]. In addition to the environment, the main factors affecting the distribution of the species include various biological and nonbiological factors, such as competition, disease, and human disturbances [49].

In contrast to the results of ecological niche modeling studies for other species [50,51], the modeling results for *V. peregusna* in this study did not show a clear pattern of response to different climatic scenarios, i.e., no shift of suitable habitats to higher latitudes or a significant decrease in the area of suitable habitats with increasing temperature. Surprisingly, the moderate carbon emission concentration at the end of this century is projected to result in the largest decrease in the suitable area for *V. peregusna*. We speculate that the reason for this result is directly related to the unique living habits of *V. peregusna* and the selection of environmental factors retained in the model. *V. peregusna* is the only mustelid animal that hibernates [25], and its food sources are predominantly rodents and lizards. Precipitation and soil factors accounted for more than 80% of the environmental influence in the modeling process, and temperature parameters, which were most influenced by changes in climate scenarios, accounted for only 0.66% of the influence in the ensemble model. Although it did not have a direct impact, climate change still had a substantial effect on the suitable habitats of *V. peregusna*, indicating that current wildlife conservation efforts face serious challenges and that it is necessary to pay increased attention to climate change to avoid species extinction.

To enhance biodiversity conservation, we recommend the following two measures to ensure that *V. peregusna* will not become an endangered species in the future. First, for regions with low climate impacts, the protection of *V. peregusna* should be strengthened, human interference and the use of anticoagulant rodenticides should be reduced, in situ protection should be enhanced, nature reserves should be established as soon as possible, and the hunting and trading of wild animals should be closely monitored. Second, for regions with relatively fragile climates, it is necessary to strengthen captive management, tentatively adopt ex situ conservation measures, promote domestication and breeding, and implement overall population resource monitoring.

The IUCN Red List is the most widely used wildlife conservation standard and reference for prioritizing conservation and ecological research [52], and the accuracy of Red List assessments has global implications [53]. IUCN assessments primarily consider the current drivers of species declines, such as population fluctuations and human pressure on populations and their suitable habitats, without adequately identifying potential future risks, such as threats posed by climate change [54,55]. Combined with studies of other listed species [56,57,58], we suggest that the IUCN consider the threats posed by climate change in future assessment efforts and incorporate distribution changes resulting from climate change into the assessment metrics for Red List species.

With the ensemble model, we effectively mitigated overfitting and improved the accuracy of predictions, but it is undeniable that the use of models to study species distributions has certain limitations. First, the results of the model are species distribution predictions, not the actual distribution of a species [59]. Second, the reproduction and migration of species are complex and dynamic processes. Natural disasters, human activities, and intraspecific competition are also important factors that influence species distributions. No existing prediction model can capture interspecies competition well. In this study, we only considered environmental factors, and this limitation may have affected the prediction accuracy to a certain extent. Third, too sparse a sample size can lead to errors in the prediction results of ecological niche models; therefore, further global co-operation is required to improve the ability to assess and protect wildlife through data sharing.

## 5. Conclusions

This study applied the ensemble model to evaluate the spatial distribution of the potential habitat of the endangered species *V. peregusna* in Eurasia and the effects of future climate change on its habitat suitability. The results showed that GBM is the best single model for predicting suitable habitats for *V. peregusna*, and the ensemble model showed higher accuracy and stability than single models. Under current conditions, the area of suitable habitats for *V. peregusna* has reached 3935.92 × 10^4^ km^2^, making it a species with a wide distribution range. In the future, climate change will severely affect the distribution and substantially reduce the area of suitable habitats for *V. peregusna*, thus posing an extremely serious challenge to the conservation of *V. peregusna*. These findings are expected to support the development of practical solutions to prevent the extinction of *V. peregusna* populations.

## Figures and Tables

**Figure 1 animals-13-02341-f001:**
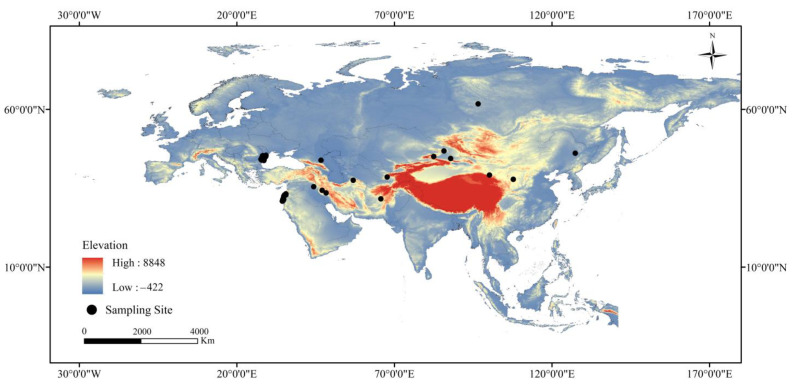
Distribution of records of *V. peregusna*.

**Figure 2 animals-13-02341-f002:**
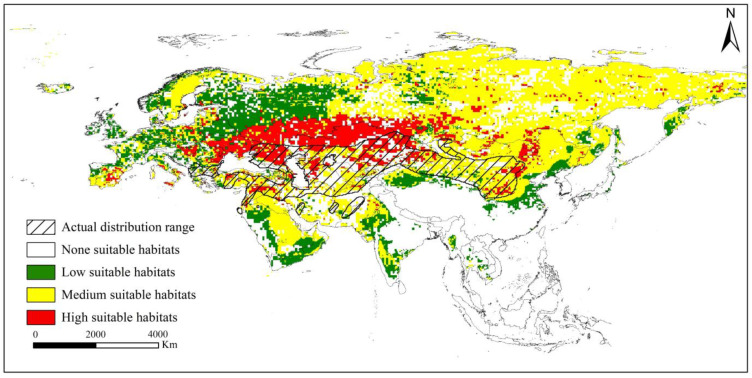
The current suitable distribution range of *V. peregusna*.

**Figure 3 animals-13-02341-f003:**
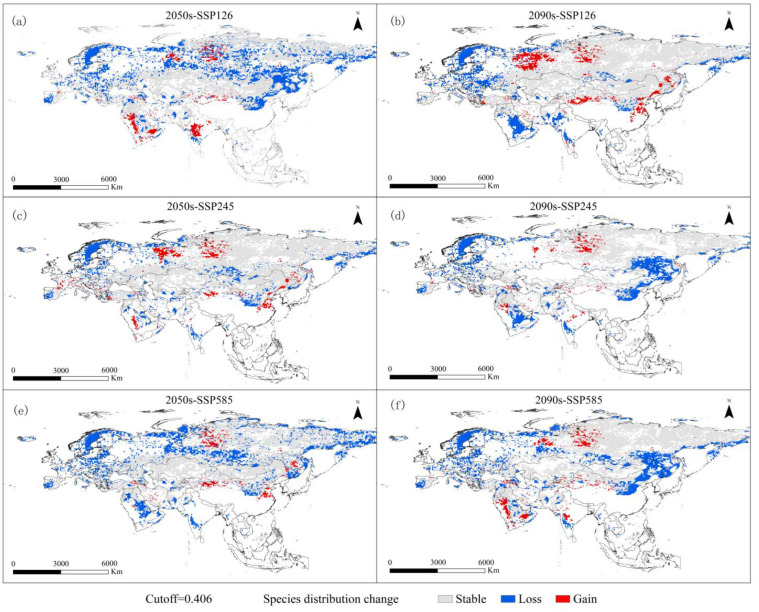
The spatial pattern changes in potential moderately and highly suitable areas for *V. peregusna* in different periods.

**Table 1 animals-13-02341-t001:** Environmental variables with their contributions and suitable value ranges.

Code	Environmental Variable	Variable Importance
bio16	Precipitation in wettest quarter	23.34
t_caco3	Topsoil calcium carbonate content	14.46
bio18	Precipitation in warmest quarter	13.94
t_teb	Topsoil teb.	12.36
bio3	Isothermality	10.65
elev	Elevation	7.27
t_cec_clay	Topsoil CEC (CLAY)	4.77
bio19	Precipitation in coldest quarter	4.26
bio17	Precipitation in driest quarter	2.18
bio15	Precipitation seasonality	2.04
t_caco4	Topsoil gypsum content	1.00
slope	Slope	0.99
bio5	Max. temperature	0.66
t_ece	Topsoil salinity (Elco)	0.56
t_gravel	Topsoil gravel content	0.43
t_oc	Topsoil organic carbon	0.41
t_esp	Topsoil sodicity (ESP)	0.37
t-sand	Topsoil sand fraction	0.30

**Table 2 animals-13-02341-t002:** Changes in moderately and highly suitable areas for *V. peregusna* in different climate scenarios in the future.

Periods	Climate Scenario	Suitable Habitat Area (×10^4^ km^2^)	Loss (×10^4^ km^2^)	Stable (×10^4^ km^2^)	Gain (×10^4^ km^2^)	Species Range Change (%)	Percentage Loss (%)	Percentage Gain (%)
Current		3067.93						
2050	SSP126	2226.43	978.64	2089.29	137.14	−27.43	31.90	4.47
	SSP245	2553.79	646.18	2421.75	132.04	−16.76	21.06	4.30
	SSP585	2226.66	926.29	2141.64	85.02	−27.42	30.19	2.77
2090	SSP126	2702.43	546.16	2521.77	180.66	−11.91	17.80	5.89
	SSP245	2038.78	1099.21	1968.72	70.06	−33.55	35.83	2.28
	SSP585	2213.48	978.59	2089.34	124.14	−27.85	31.90	4.05

## Data Availability

The original contributions presented in this study are included in the article/Appendix A, and further inquiries can be directed to the corresponding author.

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
