# Peer review of "Time to Step Up Conservation: Climate Change Will Further Reduce the Suitable Habitats for the Vulnerable Species Marbled Polecat (Vormela peregusna)"

_animals, 2023, doi:10.3390/ani13142341_

Round 1

Reviewer 1 Report

Animals-2468882

General Comments

This paper describes an ensemble species distirubion model used to predict changes to the distribution of Vormela peregusna habitat under a series of climate change scenarios. Whereas it is an important endeavor for effective conservation of the species, several changes to the work are needed before the manuscript is ready for publication.

The manuscript would be improved by including a consideration of the effect of climate changes on vegetation, which in turn should affect the habitat quality (prey density, travel corridors) of Vormela peregusna. As the manuscript is constructed now, the authors are asserting that climate affects Vormela peregusna directly, presumably due to thermal and precipitation limits. If there is some reason for this assertion, it needs to be stated in the introduction when describing the species. Alternatively, it could be that the authors have simply modeled the climate conditions where V. peregusna observed without knowing whether those conditions are at all causing V. peregusna occurrence. This limitation reduces confidence in the manuscript conclusions about habitat loss.

More information concerning the hypotheses about what drives the occurrence of Vormela peregusna and predictions for analysis are needed in the introduction and description of the methods. Without these details, the work appears to be fishing for any correlation between the occurrence data and whatever landscape/climate data can be downloaded. What do the authors expect will be important drivers of occurrence? What are the known drivers of Vormela peregusna population decline? 

Inline Comments

Abstract: the abstract requires more detail in terms of the type of model that was fit and tested. The abstract indicates that up to 33.5% of highly suitable habitat is lost, but there is no information on how habitat was defined. What types of coefficients are in the model or how the ensemble model which was undefined predicts suitable habitat.

25: be more specific. What is meant by ensemble model?

25: the first sentence says that vermila paragonza is threatened due to human loss and human threats. It is  unclear why the analysis goes on to evaluate current and future climate change. What are the tested scenarios based on? Why does climate even play a role in conservation of V. peregusna?

50: This paragraph would benefit from an explanation of what a single SDM is, rather than simply listing its uses. How do they account for landscape variation, what data are used?

61: what are the threats to V. peregusna? Why is its survival at risk? How do we know it is at risk? Who is monitoring their populations? Given what was said above, the known or predicted role of habitat fragmentation and climate change should be stated explicitly here.

104: More information on why future climates are included is needed here. Why is future climate of interest? What are the predictions about its effect?
Section 2.2 - why were no vegetation or land cover variables used? This seems to be a strange omission. More information on what is expected to drive occurrence is needed.

131: Why 1200? To match occurrence records?

147: What does this mean? How does ‘running the model’ indicate what the classifier cutoff should be?

158: The results section needs a detailed description of the importance of the model predictors. 

159: How well did the selected top or ensemble model predict the original data? Knowing this will put the next section in context, for which it would be good to know how much of the predicted habitat is actually occupied by V. peregusna.

241: The strength of the soil factors indicates that vegetation and other not-modelled covariates are likely important predictors of V. peregusna distribution and that habitat quality may not be overly sensitive to climate change.

The English is adequate.

Author Response

This paper describes an ensemble species distirubion model used to predict changes to the distribution of Vormela peregusna habitat under a series of climate change scenarios. Whereas it is an important endeavor for effective conservation of the species, several changes to the work are needed before the manuscript is ready for publication. The manuscript would be improved by including a consideration of the effect of climate changes on vegetation, which in turn should affect the habitat quality (prey density, travel corridors) of Vormela peregusna. As the manuscript is constructed now, the authors are asserting that climate affects Vormela peregusna directly, presumably due to thermal and precipitation limits. If there is some reason for this assertion, it needs to be stated in the introduction when describing the species. Alternatively, it could be that the authors have simply modeled the climate conditions where V. peregusna observed without knowing whether those conditions are at all causing V. peregusna occurrence. This limitation reduces confidence in the manuscript conclusions about habitat loss. More information concerning the hypotheses about what drives the occurrence of Vormela peregusna and predictions for analysis are needed in the introduction and description of the methods. Without these details, the work appears to be fishing for any correlation between the occurrence data and whatever landscape/climate data can be downloaded. What do the authors expect will be important drivers of occurrence? What are the known drivers of Vormela peregusna population decline? Response: Thank you very much for your careful review of our manuscript, and your valuable comments are important for improving our paper. By reviewing Vormela peregusna habits and food sources, we found that V. peregusna is an extremely widespread wildlife species, with observations of V. peregusna occurring in agricultural fields, shrubs, deserts, and hills, and therefore, we hypothesize that plants may not be the main factor influencing V. peregusna habitat The main factor of V. peregusna habitat selection. Combined with our infrared camera observations in northwestern China over the past three years, we suggest that there is a direct relationship between V. peregusna habitat selection and its main food source, rodents, especially because the activity rhythms of V. peregusna largely coincide with those of Rhombomys opimus. In earlier studies, we have tentatively demonstrated that climate change can have a large impact on R. opimus, and that the resulting changes will inevitably have an impact on V. peregusna. Unfortunately, due to the extreme lack of observations of both species, it is not possible to establish the interrelationship between the two species as their distribution points are currently defined only by a small amount of visual evidence. Therefore, in this study, we also selected environmental factors that have a strong influence on the distribution of R. opimus to analyze the fitness zone of V. peregusna. We have added the above by 2 paragraphs, in the Introduction, and thank you again for your valuable suggestions.

Inline Comments Abstract: the abstract requires more detail in terms of the type of model that was fit and tested. The abstract indicates that up to 33.5% of highly suitable habitat is lost, but there is no information on how habitat was defined. What types of coefficients are in the model or how the ensemble model which was undefined predicts suitable habitat. Response: Thanks for your comments, here we are using the ecological niche model to obtain the environmental similarity to the current distribution site to determine the potential distribution of V. peregusna. The specific parameter used is 0/1 probability value, which has been added in the modified abstract.

25: be more specific. What is meant by ensemble model? Response: Thanks for your advice, ensemble modeling refers to fitting multiple ecological niche models to achieve the best predictions, and here we used 10 models, which have been described in the abstract.

25: the first sentence says that vermila paragonza is threatened due to human loss and human threats. It is unclear why the analysis goes on to evaluate current and future climate change. What are the tested scenarios based on? Why does climate even play a role in conservation of V. peregusna? Response: In this study, we used three climate scenarios for the mid-century (2050s) and the end of the century (2090s) provided by the Coupled Model Intercomparison Project Phase 6 (CMIP6). In conjunction with your Main Comments we have added the climate change impacts on Vormela peregusna in the introduction, in addition to clarifying the future Vormela peregusna suitable areas to facilitate the establishment of protected areas and the introduction of breeding.

50: This paragraph would benefit from an explanation of what a single SDM is, rather than simply listing its uses. How do they account for landscape variation, what data are used? Response: Thank you very much for your valuable opinions. Based on your suggestion, we have added to the model introduction the method of running SDMs and the main limitations at present.

61: what are the threats to V. peregusna? Why is its survival at risk? How do we know it is at risk? Who is monitoring their populations? Given what was said above, the known or predicted role of habitat fragmentation and climate change should be stated explicitly here. Response: In conjunction with your previous comments, we have added a paragraph on the main threats to V. peregusna's survival at present and, in the context of the full text, on the possible impact of climate change on its survival.

104: More information on why future climates are included is needed here. Why is future climate of interest? What are the predictions about its effect?

Section 2.2 - why were no vegetation or land cover variables used? This seems to be a strange omission. More information on what is expected to drive occurrence is needed. Response: As we mentioned earlier, in combination with V. peregusna habits, we speculate that rodents may be the main factor influencing the distribution of V. peregusna. In our earlier studies, the effect of environment on the distribution of R. opimus has been tentatively confirmed, so we also used consistent environmental data in this study. We strongly acknowledge that the vegetation conditions and land use type factors you mentioned may be the main factors influencing the distribution of V. peregusna within a small environment, but there are great practical difficulties in the current modeling, partly because V. peregusna is a widely distributed species across Asia and Europe, and the current data on vegetation and land use types are mostly country or region based studies, and there are problems in building models with such a wide range of Practical problems. On the other hand, vegetation distribution is also influenced by climate and environment, and data about soil aspects are currently used in the model. There are also some cases here where similar models were used, and all of them used environmental factors that are consistent with this study. Gebrewahid, Y., Abrehe, S., Meresa, E. et al. Current and future predicting potential areas of Oxytenanthera abyssinica (A. Richard) using MaxEnt model under climate change in Northern Ethiopia. Ecol Process 9, 6 (2020). https://doi.org/10.1186/s13717-019-0210-8 Khan, A.M.; Li, Q.; Saqib, Z.; Khan, N.; Habib, T.; Khalid, N.; Majeed, M.; Tariq, A. MaxEnt Modelling and Impact of Climate Change on Habitat Suitability Variations of Economically Important Chilgoza Pine (Pinus gerardiana Wall.) in South Asia. Forests 2022, 13, 715. https://doi.org/10.3390/f13050715

131: Why 1200? To match occurrence records? Response: The choice of 1200 pseudoabsence is the basic condition for modeling, which needs to satisfy more than 10 times of the actual distribution of points, and we actually collected a total of 101 points that were de-weighted within 100 km, so this value was chosen.

147: What does this mean? How does ‘running the model’ indicate what the classifier cutoff should be? Response: Since we are using the Biomod2 platform, this provides us with a number of simplified operations for building the integrated model. When the model testing is complete, typing "running the model" loads the get_evaluations(myBiomodModelOut) function within biomod to calculate the environmental thresholds for the areas where the test set is most similar to the environmental conditions of the currently provided points, i.e., cut-off, and since biomod provides simplified operations, this section is provided to allow other researchers to replicate our research.

158: The results section needs a detailed description of the importance of the model predictors. Response: Thank you very much for your suggestion. Regarding the importance of the modeling factors we have provided a table in the original manuscript in section 2.2 (Table1). The importance of all participating modeling factors is represented in the form of weights.

159: How well did the selected top or ensemble model predict the original data? Knowing this will put the next section in context, for which it would be good to know how much of the predicted habitat is actually occupied by V. peregusna. Response: We used all 3 metrics currently validated by the ecological niche model: AUC, KAPPA, and TSS, and all 3 metrics were greater than 0.9, showing that our prediction accuracy was extremely high, as has been expressed in section 3.1 of the original manuscript. Since we also tested all 3 metrics for all single models, this section would have taken up a tremendous amount of space if presented in the main text, so we have presented the results for all single models through supplementary materials. Regarding the current distribution you mentioned, unfortunately V. peregusna is an extremely rare and endangered species, and it is difficult for us to specify its current true range; all data are from V. peregusna sightings, so we cannot obtain the area of its current range to compare with the model results. Here, we have combined your and another reviewer's comments and added the existing sighting area provided by IUCN to Fig.2, but since the distribution area provided by IUCN does not contain vector information, we cannot calculate its area and can only represent it by in-plot annotation.

241: The strength of the soil factors indicates that vegetation and other not-modelled covariates are likely important predictors of V. peregusna distribution and that habitat quality may not be overly sensitive to climate change. Response: From a short-term and small-scale ecological area, vegetation and other factors may be one aspect of the cause of the distribution of V. peregusna. Combining the results we have found and the current main habits of V. peregusna, both vegetation and other factors will receive climate change effects on a longer predicted time. As in the Response to the previous questions, this part has been added in the Introduction. Once again, we sincerely thank you for your contribution to the improvement of our manuscripts.

Reviewer 2 Report

Dear authors, 

Thanks for reviewing this interesting paper. I think this paper can have importance for detecting susceptibility of many species in relation to predicting effects of climate change on the possibility of future distribution of many species. Especially those species of mammals, birds or insects living in marginal habitats or for example species living in mountain areas where the climate change will lead to that the adaptability to alpine conditions, will force these species to move to higher altitudes and have less available habitat. Modeling species possible distribution in the future will always be important. 

Dear authors, 

Thanks for reviewing this interesting paper. I think this paper can have importance for detecting susceptibility of many species in relation to predicting effects of climate change on the possibility of future distribution of many species. Especially those species of mammals, birds or insects living in marginal habitats or for example species living in mountain areas where the climate change will lead to that the adaptability to alpine conditions, will force these species to move to higher altitudes and have less available habitat. 

Author Response

Thank you very much for your approval of our manuscript and for taking the time to review it.

Reviewer 3 Report

The authors assessed the distribution of suitable habitats for the marbled polecat (Vormela peregusna) by using a predictive model to evaluate potential habitats under the current and future climate change conditions. The manuscript is well written, but there seems to be a lack of a clear distinction between the distribution of the animals and the suitable habitats for the animals. The authors need to ensure that they are clear what they are investigating, the actual distribution or the suitable habitats, and retain this distinction throughout the text.

Some specific comments:

It may be a good idea to add the common name of the species, if not in the title, at least in the abstract, since all readers may not be familiar with the scientific name.

Do we know what the relationship is between suitable areas and occurrence of the polecat in it? For example, an area may be suitable for a species, but do they actually occur there?

Line 23 – replace ‘as of now’ with ‘currently’

Line 28 – remove ‘the’ between predicting and suitable

Line 50 – remove ‘in abundance’

Line 151 – extra space after nodes

For your current suitability map (Figure 2), it may be useful to also include a map of where the animals actually occur at the moment.

Figure 2 – you need to clarify in the legend that it is current suitable habitat for the species, not the actual distribution range of the animals.

Figure 3 – what does ‘stable’ signify? Is it suitable areas that are maintained? Also, does this figure indicate the species distribution range (as indicated at the bottom of the graph), or the suitable habitats for the species, which could be vastly different?

Table 2 – do you provide the parameters for each climate scenario somewhere?

In the discussion you mention that carbon emission concentration turned out to be one of the primary drivers of the decrease in suitable areas, but this is not indicated in the results section – I suggest you add a table that indicate the various parameters included in the most suitable models and indicate somewhere in the results which factors are the most prominent.  

 Line 240 – food sources are primarily rodents and lizards in deserts – large areas of suitable habitats are located in Europe in non-desert areas, you will have to revise this statement to include food sources when the animals do not occur in deserts.

Good, some small changes are needed

Author Response

Some specific comments:

It may be a good idea to add the common name of the species, if not in the title, at least in the abstract, since all readers may not be familiar with the scientific name.

Response: Thank you very much for your suggestion. We have added the common name Marbled Polecat to both the title and the abstract of the manuscript.

Do we know what the relationship is between suitable areas and occurrence of the polecat in it? For example, an area may be suitable for a species, but do they actually occur there?

Response: The relationship between the true distribution of the species and the suitable habitats that you mention involves the basic assumption of the ecological niche model, that is, that the results derived from the model are the most environmentally similar areas within the interval fitted to the true distribution. There is no doubt that the predicted results do not mean that the species will definitively occur in this area, but for these species, the predicted area will satisfy the basic conditions for their survival. The maximum purpose of this result can be used for the identification of protected areas.

Line 23 – replace ‘as of now’ with ‘currently’

Response: Thanks to Reviewer for pointing out our mistakes, this sentence was removed due to changes made to the abstract by other reviewers.

Line 28 – remove ‘the’ between predicting and suitable

Response: We have made correction according to the Reviewer’s comments.

Line 50 – remove ‘in abundance’

Response: We have made correction according to the Reviewer’s comments.

Line 151 – extra space after nodes

Response: We have made correction according to the Reviewer’s comments.

For your current suitability map (Figure 2), it may be useful to also include a map of where the animals actually occur at the moment.

Figure 2 – you need to clarify in the legend that it is current suitable habitat for the species, not the actual distribution range of the animals.

Response: Thank you very much for your valuable suggestion, we obtained the distribution of this species from IUCN and added it to the figure, also adding the true distribution description at the legend.

Figure 3 – what does ‘stable’ signify? Is it suitable areas that are maintained? Also, does this figure indicate the species distribution range (as indicated at the bottom of the graph), or the suitable habitats for the species, which could be vastly different?

Response: Stable refers to areas that do not change in the future. Since this species is an extremely rare species and no country or organization has conducted monitoring for this population, all areas in our manuscript are fitted potential results from sighting evidence, and the figure captions have been modified.

Table 2 – do you provide the parameters for each climate scenario somewhere?

In the discussion you mention that carbon emission concentration turned out to be one of the primary drivers of the decrease in suitable areas, but this is not indicated in the results section – I suggest you add a table that indicate the various parameters included in the most suitable models and indicate somewhere in the results which factors are the most prominent.  

Response: Thank you very much for your valuable comments, we use here the future climate scenarios under 3 carbon emission models for 2 future periods provided by CIMP6, because these climate scenarios involve many complex meteorological parameters, so it is difficult to show the differences in the form of a table, in addition, this part has been published in many meteorological papers, the specific content can be accessed through these papers, adding here may cause duplication of content.

 Line 240 – food sources are primarily rodents and lizards in deserts – large areas of suitable habitats are located in Europe in non-desert areas, you will have to revise this statement to include food sources when the animals do not occur in deserts.

Response: Thanks to your suggestion, we removed the word desert to ensure an accurate description.

Round 2

Reviewer 1 Report

The manuscript has been improved.    The explanation for not including vegetation cover is adequate. The assertion that the most important driver of V. peregusna distribution is prey, is not being tested in their work. However, using climate and landcover (soil cover in this) case is a typical proxy in this kind of work.      

Author Response

Thank you very much for reviewing our manuscript again, your valuable comments are very helpful to improve this manuscript and our future research. Regarding the question of whether food is the main factor influencing the distribution of V. peregusna, unfortunately, because both rodents and V. peregusna are extremely difficult to observe, there is still no clear conclusion, and this part of the speculation is also combined with the "More information concerning the hypotheses about what drives the occurrence of Vormela peregusna and predictions for analysis are needed in the introduction and description of the methods" mentioned by your first revision. We have added in the Introduction that this speculation is based on speculation and is not real. (L85-86)

As you mentioned, using a combination of climate change and land use to predict species distributions would further improve the accuracy of our assessments. However, currently this method is mostly used in precision planting of plants, and there are some difficulties in applying this parameter in this manuscript. On the one hand, land use-related studies are more often used at small scales, and it is difficult to obtain specific data at large regional scales such as Eurasia. On the other hand it is that the focus of this manuscript is to analyze future changes in the fitness zones of V. peregusna, whereas future land use is more relevant to the policies of different governments and is suitable for the analysis of differences between the past and the present, which is difficult to apply to future changes in the fitness zones (Dhami et al.,2023; Koli et al.,2023; Adhikari et al. 2023).

We sincerely thank you for your great contribution to our manuscript.

Adhikari, Binaya, Subedi, Suresh C., Bhandari, Shivish, Baral, Kedar, Lamichhane, Sandesh, and Maraseni, Tek.  “ Climate-Driven Decline in the Habitat of the Endemic Spiny Babbler (Turdoides Nipalensis).” Ecosphere, 2023, 14( 6): e4584.

Dhami, B.; Bhusal, A.; Adhikari, B.; Miya, M.S.; Maharjan, S.K.; Neupane, D.; Adhikari, H. Habitat Suitability and Conflict Zone Mapping for the Blue Bull (Boselaphus tragocamelus) across Nepal. Animals 2023, 13, 937.

Koli, V.K.; Jangid, A.K.; Singh, C.P. Habitat suitability mapping of the Indian giant flying squirrel (Petaurista philippensis Elliot, 1839) in India with ensemble modeling. Acta Ecologica Sinica, 2023, 43, 644-652.